# Current Options and Future Perspectives in the Treatment of Dyslipidemia

**DOI:** 10.3390/jcm11164716

**Published:** 2022-08-12

**Authors:** Saverio Muscoli, Mihaela Ifrim, Massimo Russo, Francesco Candido, Angela Sanseviero, Marialucia Milite, Marco Di Luozzo, Massimo Marchei, Giuseppe Massimo Sangiorgi

**Affiliations:** 1Division of Cardiology, “Tor Vergata” University Hospital, v.le Oxford 81, 00133 Rome, Italy; 2Department of Biomedicine and Prevention, “Tor Vergata” University, 00133 Rome, Italy

**Keywords:** dyslipidemia, PCSK9 inhibitors, bempedoic acid, inclisiran, LXR agonists, PPARβ/δ, ANGPTL3 inhibitors, mipomersen, lomitapide, volanesorsen

## Abstract

Low-density lipoprotein cholesterol (LDL-C) plays a crucial role in the development of atherosclerosis. Statin therapy is the standard treatment for lowering LDL-C in primary and secondary prevention. However, some patients do not reach optimal LDL-C target levels or do not tolerate statins, especially when taking high doses long-term. Combining statins with different therapeutic approaches and testing other new drugs is the future key to reducing the burden of cardiovascular disease (CVD). Recently, several new cholesterol-lowering drugs have been developed and approved; others are promising results, enriching the pharmacological armamentarium beyond statins. Triglycerides also play an important role in the development of CVD; new therapeutic approaches are also very promising for their treatment. Familial hypercholesterolemia (FH) can lead to CVD early in life. These patients respond poorly to conventional therapies. Recently, however, new and promising pharmacological strategies have become available. This narrative review provides an overview of the new drugs for the treatment of dyslipidemia, their current status, ongoing clinical or preclinical trials, and their prospects. We also discuss the new alternative therapies for the treatment of dyslipidemia and their relevance to practice.

## 1. Introduction

The main cause of death worldwide is cardiovascular disease (CVD). The number of deaths from CVD has increased exponentially over the last decades, from 2 million in 2000 to almost 9 million in 2019. In fact, 32% of all global deaths in 2019 were caused by CVD, with 17.9 million people dying from these diseases in absolute terms [1]. 

Dyslipidemia, together with diabetes, is one of the major risk factors for CVD. Several clinical studies have demonstrated the role of low-density lipoprotein cholesterol (LDL-C) in the development of atherosclerosis and CVD and a proportional relationship between its lowering and the decrease in acute myocardial infarction (AMI) and chronic coronary syndrome (CCS), leading to the motto “the lower the better” [2].

Over the last 30 years, there has been a whole generation of clinical trials demonstrating a reduction in cardiovascular (CV) risk through therapies with increasingly intensive lipid-lowering agents, such as statins in combination with other agents, such as ezetimibe [3]. The standard of care for LDL-C-lowering is statin, which reduces CV risk by 25–35% [3]. There is wide evidence of statins’ efficacy in lowering LDL-C and simultaneously lowering the risk of CV events in both primary and secondary prevention. 

According to meta-analysis of 14 randomized trials (CARE, AFCAPS/TexCAPS, 4S, PROSPER, WOSCOPS, Post-CABG, LIPS, LIPID, GISSI Prevention, HPS, ASCOT-LLA, ALLHAT-LLT, CARDS, ALERT), statin therapy leads to a decrease of up to 12% in all-cause mortality per mmol/L LDL-C lowered [4]. 

From a pharmacological perspective, statins reduce intrahepatic production of cholesterol by competitively inhibiting a key enzyme, 3-hydroxy-3-methylglutaryl-coenzyme A (HMG-CoA) reductase, thereby limiting the conversion of HMG-CoA to mevalonic acid, a precursor of cholesterol. The reduction of hepatic cholesterol synthesis leads to overexpression of LDL receptors (LDL-R) and consequently to increased uptake of cholesterol into the cells, resulting in a decrease in plasma cholesterol levels [2]. Nevertheless, some patients treated with high-dose statins fail to reach LDL-C targets. Other patients experience statin intolerance (SI) or side effects, such as myalgia, rhabdomyolysis, cramps, and weakness [2]. Millions of treated patients are statin intolerant; they may not tolerate the full therapeutic statin dose and have adverse effects, mostly myopathy.

Currently, there are no reliable data on the prevalence of SI; this can vary by up to 5–70% from different studies. Such a sizeable interindividual variation is attributed, at least partly, to polymorphisms in the genes that influence the pharmacodynamics and pharmacokinetics of statins [5].

Because a significant number of CV events persist despite maximal therapy, it has been necessary to combine different therapeutic approaches and test several new drugs to reduce the burden of CVD [3]. Research has therefore focused on the development of alternative therapies that, alone or in combination with statins, effectively and durably reduce LDL-C levels, while being well tolerated and having a rather harmless side effect profile [2].

These strategies include relatively new agents, such as proprotein convertase subtilisin/kexin type 9 (PCSK9i) and bempedoic acid (BA), which are currently used in clinical practice. Nowadays, the novel role of these new strategies in lowering LDL-C and triglycerides is being further investigated. 

Another severe disease is familial hypercholesterolemia (FH), a codominant monogenic dyslipidemia that can lead to CV events early in life, especially in the homozygous form (HoFH) [5]. Its heterozygous form (HeFH) has an estimated prevalence of 1/200–250, while HoFH is a rare disease with a prevalence of 1 in 160,000–320,000. HoFH is a major challenge for physicians because these patients respond poorly or not to conventional pharmacological treatment, and until a few years ago lipoprotein apheresis was the only solution. Recently, new and promising pharmacological strategies have also become available for HoFH. [6]. This disease is a major challenge for physicians because these patients respond poorly or not at all to conventional pharmacological treatment, and until a few years ago, lipoprotein apheresis was the only solution. Recently, new and promising pharmacological strategies have also become available for HoFH [6].

In this narrative review, we provide an overview of the new drugs used in the treatment of dyslipidemia, their results in the clinical arena, current preclinical or clinical studies, and future prospects.

## 2. PCSK9 Inhibitors

PCSK9 is a serine protease found in many tissues but mainly expressed in the liver that targets LDL-R. It leads the receptors to lysosome-mediated degradation, thus diminishing their recycling and decreasing the removal rate of circulating LDL-C with a subsequent increase in LDL-C concentration in the blood [2,7,8,9] (Figure 1). 

Therefore, two monoclonal antibody inhibitors of PCSK9 were developed, and approved by the US Food and Drug Administration (FDA) in 2015 as a therapeutic option in addition to diet and a maximally tolerated dose of statins for patients with FH and in cases of SI. These newly approved antibodies are alirocumab and evolocumab. In accordance with the current guidelines of the American Heart Association (AHA) [10] and European Society of Cardiology (ESC) [11], they may be used alone or in combination with statins and ezetimibe to achieve therapeutic goals. PCSK9i have been shown to markedly reduce LDL-C and are characterized by a low incidence of adverse events (ADEs) as demonstrated in the FOURIER and ODYSSEY trials [12,13]. 

The ODYSSEY LONG TERM is a randomized, double-blind, controlled trial that investigated the effects of 78 weeks of therapy with alirocumab vs. placebo in 2341 patients at high risk for CVD who were already receiving the maximum tolerated dose of statins [13]. Compared with placebo, alirocumab led to a reduction of 61.9% of LDL-C. Furthermore, patients taking alirocumab, as compared with placebo, also observed reduced levels of total cholesterol, non–high-density lipoprotein cholesterol (HDL-C), apolipoprotein B (ApoB), lipoprotein(a) (Lp(a)) and fasting triglycerides by 37.5%, 52.3%, 54%, 25.6% and 17.3%, respectively and augmented levels of ApoA1 and HDL-C and by 2.9% and 4.6% respectively [13,14].

Moreover, PCSK9i have proved effective in patients with SI. In a clinical study, patients at moderate-to-high CV risk with SI were randomized either to alirocumab treatment or ezetimibe treatment, resulting in a mean LDL-C reduction of 45.0% with alirocumab and a mean LDL-C reduction of 14.6% with ezetimibe (mean difference 30.4%, *p* < 0.0001) [15]. 

The ODYSSEY OUTCOMES, a multicenter, double-blind, randomized trial, enrolled 18,924 patients with acute coronary syndrome and an LDL-C level of ≥1.81 mmol/L (70 mg/dL), a non-HDL-C level of ≥2.58 mmol/L (100 mg/dL) or an ApoB level of ≥1.56 μmol/L (80 mg/dL). All patients were treated with the maximum tolerated statin dose. Alirocumab reduced the primary endpoint (composite of death from coronary artery disease, non-fatal AMI, fatal or non-fatal ischemic stroke or unstable angina requiring hospitalization) and deaths from any cause by 15% over a median follow-up period of 2.8 years [16]. 

The FOURIER trial, a double-blind, randomized, placebo-controlled trial enrolled 27,564 patients with stable atherosclerotic cardiovascular disease (ASCVD) and additional risk factors. Treatment with evolocumab reduced the trial’s primary endpoint (composite of CV death, AMI, stroke, coronary revascularization or hospital admission for unstable angina) by 15%. In addition, it reduced the key secondary endpoint (composite of CV death, AMI or stroke) by 20% over an average follow-up period of 2.2 years. The results had not shown variations across subgroups, and the effect of the therapy appeared to increase over time [17]. 

Considering the substantial reduction in LDL-C concentration observed with PCSK9 inhibition, concerns also arose regarding potential safety issues. A pre-specified secondary analysis of the FOURIER study addressed this issue and showed a strong relationship between the LDL-C level achieved and key CV outcomes up to LDL-C levels < 0.2 mmol/L (7.7 mg/dL). These data argued for more aggressive lowering of LDL-C in patients with CVD to levels well below current recommendations and led to the dictum in the medical community that “the lower the better” [18]. 

A raising concern after earlier trials of PCSK9i was the higher incidence of adverse neurocognitive events in the group of patients treated with PCSK9i as compared with placebo [12,13]. However, the subsequent FOURIER [17] and ODYSSEY OUTCOMES [16] clinical trials did not reveal a significant difference in the incidence of adverse neurocognitive events between patients treated with a PCSK9i versus placebo. Therefore, the recent EBBINGHAUS study, a double-blind, randomized, placebo-controlled study, specifically investigated this issue. A total of 1204 patients from the FOURIER study was enrolled and their cognitive functions were prospectively assessed with the Cambridge Neuropsychological Test Automated Battery. 

The EBBINGHAUS study showed that there were no significant differences in cognitive function over a median of 19 months between patients who received evolocumab or placebo in addition to statin therapy. Furthermore, no association between LDL-C levels and cognitive changes was found in this study [19]. 

PCSK9i adverse effects of are usually mild and include upper respiratory tract infections, injection-site reactions and nasopharyngitis [20,21].

In addition, in other studies, PCSK9i has been shown to reduce the lipid core of atherosclerotic plaques. In this setting, carotid plaques were evaluated with MRI after six months of PCSK9i with alirocumab, and it resulted in reduced lipid content by 17%, without significant changes in lumen/wall area or in the inflammatory index Ktrans [22]. 

Lepor et al. analyzed carotid plaques with MRI after 3, 6 and 12 months of therapy with PCSK9i, founding a regression in plaque composition and neovasculature [23]. Similar findings of morphological stabilization and reduction of carotid plaques were reported in the CARUSO study after 6 months of treatment with evolocumab on top of ongoing lipid-lowering therapy [24].

In addition, the subanalysis of FOURIER reported that the addition of evolocumab to standard therapy significantly decreased the risk of developing complex coronary disease requiring revascularization [25]. 

PACMAN-AMI A double-blind, placebo-controlled study RCT investigated the effect of early treatment with alirocumab in patients with AMI, in whom serial multimodal intracoronary imaging (IVUS, NIRS and OCT) evaluated non culprit high-risk lesions. Early treatment with alirocumab, in addition to high-dose statin therapy, resulted in significantly greater regression of coronary plaque in non-infarct-prone lesions after 52 weeks compared to placebo [26]. 

These trials provide unique insight into the biological process of plaque stabilization with disease-modifying therapies and shed light on new perspectives.

According to the current European and American guidelines, PCSK9i have become a current treatment in managing dyslipidemia both in primary and secondary prevention on top of standard therapy [10,11].

## 3. Bempedoic Acid

The enzyme acyl-CoA synthetase 1 (ACSVL1), expressed primarily in the liver, converts the prodrug BA into the active metabolite (ETC-1002-CoA). After activation, BA inhibits adenosine triphosphate citrate lyase, resulting in a reduction in acetyl-CoA levels at a level in the cholesterol synthesis pathway upstream of HMG-CoA reductase, the molecular target of statins (Figure 1). This results in a decrease cholesterol synthesis, thus leading to an upregulation of LDL-R and a subsequent lowering of LDL-C levels [2].

In addition, activation of AMP-activated protein kinase leads to inhibitory phosphorylation of HMG-CoA reductase and acetyl-CoA carboxylase, improves glucose regulation [27], and reduces proinflammatory cytokines production chemokines in human macrophages [28]. Furthermore, the skeletal muscle cannot activate prodrug due to the absence of the enzyme ACSVL1, resulting in the reduction of adverse muscle effects, which often complicate statin therapy [29,30].

BA is a novel once-daily oral molecule with a half-life of 15–24 h. The first phase 2 trial conducted by Ballantyne et al. was a 12-weeks study of 177 patients, 133 of whom were to investigate the LDL-C reduction efficacy and safety of BA compared with placebo in patients with hypercholesterolemia (LDL-C of 3.36 to 5.68 mmol/L (130 to 220 mg/dL)) and normal or elevated (>1.69 mmol/L (150 mg/dL)) triglycerides [27]. LDL-C levels were significantly reduced by an average of 18, 25 and 27% at doses of 40, 80 and 120 mg, respectively, compared with an average of 2% in patients treated with placebo. BA also lowered atherogenic biomarkers ApoB, non-HDL-C and LDL particles. Gutierrez et al. showed that in 60 patients with type 2 diabetes, BA (80 mg once daily for 2 weeks, followed by 120 mg once daily for 2 more weeks) lowered LDL-C and high-sensitivity C-reactive protein (hs-CRP) (−43% and −41%, respectively) compared with placebo (−4% and −11%) [31]. 

Ballantyne et al. studied the efficacy of BA (120 mg and 180 mg) or placebo, in addition to ongoing statin therapy, in 134 patients with hypercholesterolemia (LDL-C of 2.97–5.68 mmol/L (115–220 mg/dL)) for 12 weeks [32]. 

BA lowered LDL-C levels by up to 24% more than placebo (−4.2%); it also reduced hs-CRP, apoB, non-HDL-C and total cholesterol levels more than placebo. No significant reductions in HDL-C or triglyceride levels were observed. When muscle-related ADEs occurred, the number of drug discontinuations and the number of clinical safety trials were generally similar to placebo. Thompson et al. compared the LDL-C-lowering efficacy of BA (120 or 180 mg) versus ezetimibe with (*n* = 177) or without (*n* = 171) SI in 348 patients. Compared to ezetimibe in monotherapy, a significantly greater reduction in LDL-C of up to 30% was observed in patients receiving BA. In patients receiving BA and ezetimibe, LDL-C reduction of up to 48% was observed. Moreover, a hs-CRP reduction of 40% with BA was detected, and there was no increase in muscle-related ADEs compared to ezetimibe [33]. 

The CLEAR Harmony was a multicenter phase 3 trial, including 2230 patients with ASCVD and/or HeFH, LDL-C > 1.81 mmol/L (70 mg/dL) whilst on treatment with a maximum tolerated dose of statin randomized for 52 weeks. The patients were randomly assigned to receive BA (180 mg) or placebo. An evaluation after 12 weeks, observed that the mean LDL-C level decreased by 18.1% compared to placebo. No significant difference regarding the ADEs was found between the two groups. However, BA led to a higher incidence of ADEs leading to discontinuation and gout [34].

CLEAR Wisdom phase 3 trial evaluated the reduction in LDL-C levels in patients with maximum tolerated dose of statin, LDL-C > 70 mg/dL, ASCVD and/or HeFH. Patients were randomly assigned to receive BA (180 mg) vs. placebo; a reduction of LDL-C of 15.1% was observed in BA group vs. placebo 2.4% [35]. The CLEAR Serenity a 24-weeks, randomized phase 3 trial, evaluated 345 SI patients affected by hypercholesterolemia, who required lipid-lowering therapy. The patients were randomly assigned to receive BA or a matching placebo. At week 12, in the group receiving BA there was a significant reduction of LDL-C from baseline (−21.4%) and a lower incidence of myalgia compared with placebo (4.7% versus 7.2%, respectively), without significant changes in HDL-C and triglycerides levels [36]. 

The efficacy and safety of BA in combination with ezetimibe was investigated in the CLEAR Tranquilly study (III), which included 269 participants with hypercholesterolemia (SI) who were already receiving stable lipid-lowering therapy. Following 4-weeks of ezetimibe 10 mg, patients were randomly assigned to receive in addition BA (180 mg) or placebo for 12 weeks. BA reduced LDL-C levels by 28.5% compared to placebo. The association was found to be safe and effective, with no significant difference in side effects compared to placebo [37]. 

In all CLEAR phase 3 trials, BA also lowered hs-CRP, ApoB, non-HDL-C and total cholesterol levels by more than placebo. The most frequent ADEs associated with BA therapy were urinary tract infections (4.5%), reduction in the glomerular filtrate (0.7%), headache (2.8%), hyperuricemia (2.1%) and gout (1.4%) [38]. On the other hand, BA was associated with a reduced risk of new-onset or worsening of diabetes mellitus [39]. The CLEAR outcome included 14,014 participants and is the first ongoing randomized trial to examine the effects of BA on CV events in patients with SI predisposed to or with CVD [40]. 

In conclusion, BA represents a useful addition to LDL-C lowering therapies in patients with different CV and/or HeFH risk profiles on statin therapy or intolerant to them, whose LDL-C levels remain above those suggested by the current guidelines.

## 4. Inclisiran

The use of small interfering RNA (siRNA) represents another strategy to reduce PCSK9 secretion. siRNAs block the expression of specific genes with complementary nucleotide sequences by selectively silencing the translation of their complementary target mRNAs [41]. Inclisiran (Figure 1), targeting the 3′ UTR of the PCSK9 mRNA, is a long-acting, subcutaneously delivered, synthetic siRNA conjugated to triantennary N-acetylgalactosamine carbohydrates. It binds to asialoglycoprotein receptors in the liver, resulting in uptake of Inclisiran and suppression of hepatic PCSK9 production, leading to elevation of LDL-R in hepatocyte membranes and a subsequent decrease in circulating LDL-C levels [42]. Peak plasma concentrations of Inclisiran are reached after 4 h with a plasma half-life 5–10 h, which is not influenced by kidney failure, despite having a renal elimination. After a single injection, the LDL-C effects of the drug are reversed at a rate of approximately 2% per month [43,44]. The ORION program consists of worldwide trials to evaluate the efficacy and safety of inclisiran in a particular population, including those with established ASCVD or FH and those at high risk of ASCVD [2].

The results on the efficacy of this drug are mainly based on 3 pivotal 18-month phase 3 clinical trials: ORION-9, ORION-10 and ORION-11, multicenter, randomized, double-blind, placebo-controlled clinical trials. The two primary endpoints were the seventeenth month in the LDL-C level variation and the time-adjusted variation between the third and eighteenth months from baseline. In ORION-9 482 patients (47% men; median age 56 years; 3.95 mmol/L (153 mg/dL) mean baseline LDL-C) with HeFH were randomly assigned to receive sc injections of 300 mg of inclisiran sodium or placebo on the first day, third month, the ninth month, and fifteenth month (in a 1:1 ratio). After 17 months, the LDL-C level increased by 8.2% in the placebo group while was observed a reduction of 39.7% in the inclisiran group. The time-averaged percentage variation in the LDL-C level between month 3 and month 18 increased 6.2% in the placebo group and a reduction of 38.1% in the inclisiran group. There were significant lowerings in LDL-C levels in all genotypes of FH, while serious ADEs were similar in the two groups [45].

ORION-10 trial enrolled patients with ASCVD, while the ORION-11 trial evaluated patients receiving statin therapy at the maximum tolerated dose with ASCVD or an ASCVD risk equivalent who had elevated LDL-C levels. A total of 1561 participants with LDL-C at a baseline of (2.71 ± 0.99 mmol/L (104.7 ± 38.3 mg/L)) and 1617 patients, with LDL-C levels at baseline of (2.73 ± 1.01 mmol/L (105.5 ± 39.1 mg/dL)) were randomized in the ORION-10 and 11 trials, respectively. Specifically, 284 mg of inclisiran or placebo was randomly administered (in a 1:1 ratio) by sc injection on day 1, month 3, and every 6 months after that over 18 months. After 17 months of treatment, inclisiran reduced LDL-C levels by 52.3% and 49.9% in the ORION-10 and 11 trials, respectively (*p* < 0.001 for all comparisons vs. placebo). ADEs were generally similar in the inclisiran and placebo groups in each trial, although injection-site ADEs were more frequent with inclisiran than with placebo (2.6% vs. 0.9% in the ORION-10 trial and 4.7% vs. 0.5% in the ORION-11 trial); such reactions were generally mild, and none were severe or persistent [2].

In the safety analysis of ORION-1, no ADEs related to inflammation, immune activation, or clinical immunogenicity was observed. There were no liver, kidney, or muscle toxicity signs in any trial with inclisiran. There have been no reported drug–drug interactions with inclisiran, statins, or other medications. Additionally, the safety of inclisiran has been tested in individuals with renal impairment and dose adjustments are not required in this subgroup of patients [45,46,47].

ORION-4 is a phase 3, double-blind, randomized, ongoing controlled trial evaluating the reduction of MACE in participants with pre-existing ASCVD who cannot achieve LDL-C targets. This trial is expected to involve approximately 15,000 subjects aged > 55 years, they will be randomized between inclisiran sodium 300 mg and matching placebo [48].

There are expected to be 20 ORION trials of inclisiran, five of which will involve special populations (e.g., people with impaired kidney function or liver dysfunction). ORION-3 is a phase 3, an active comparative study evaluating the efficacy, safety, and tolerability of inclisiran and evolocumab in participants at high CV risk and with elevated LDL-C. ORION-8 is an ongoing study, evaluating the long-term efficacy and safety of inclisiran in patients with ASCVD, HeFH, or HoFH [45]. Inclisiran was recently approved by the European Medicines Agency (EMA) to treat adults with primary hypercholesterolemia or mixed dyslipidemia. 

Inclisiran has shown promise as a new agent for the treatment of hypercholesterolemia. It offers infrequent, convenient, twice-yearly dosing, while providing significant, durable, LDL-C lowering, with a favorable side-effect profile. The results of ongoing trials are eagerly awaited as they will demonstrate Inclisiran’s role in reducing CV risk and improving clinical outcomes in patients with ASCVD [43].

## 5. LXR Agonists

Liver X receptors (LXRs) are transcription factors that play a key role in lipid homeostasis and reverse cholesterol transport (ReCT), the process by which HDL transports excess cholesterol from extrahepatic structures back to the liver for metabolism and excretion (Figure 1) [49,50,51]. LXRs include two isoforms, LXR α and LXR β [52]. While LXR β is ubiquitously expressed, LXR α is expressed exclusively in the lung, intestine, liver, kidney, adipose tissue, and immunocytes [53,54]. They are nuclear ligand-activated receptors; oxysterols and other cholesterol metabolites LXR binding induce a conformation change that causes corepressor releasing, coactivator recruitment, and transcription of target genes [55]. Activation of LXR in macrophages induces cholesterol efflux up-regulating the adenosine triphosphate-binding cassette (ABC) A1 and G1 [56]. Studies have shown that ABCA1 promotes cholesterol efflux by binding to apolipoprotein A-I. In addition to its effect on lipid metabolism, it plays a crucial role in anti-inflammation [57]. Moreover, LXRs reduce intestinal cholesterol absorption and promote intestinal excretion [58,59], increasing apolipoprotein E expression [60], and downregulating Niemann-Pick C1 like 1 [61]. LXRs stimulation revealed as a potential mechanism of action for the development of new encouraging therapeutic strategies [62]. Despite the anti-atherosclerotic and anti-inflammatory properties of LXR agonists, full LXR natural or synthetic agonism increase hepatic lipogenesis, leading to hypertriglyceridemia and liver steatosis. So, despite beneficial CV effects, T0901317 and GW3965 [63] have not yet been tested in clinical trials because of their adverse effects. The previous study suggests that LXR α, which is highly expressed in the liver, is mainly responsible for these adverse effects modulating the hepatic expression of sterol regulatory element-binding protein (SREBP-1) [64,65,66]. These data support the idea of limiting LXR-α activity to avoid the undesirable lipid side effects of LXR agonists and suggest targeting of LXR-β to achieve anti-atherosclerotic benefits. LXR-623 is the first LXR-targeting compound to be evaluated in clinical trials; it is a synthetic LXR modulator with high selectivity and oral bioavailability that preferentially binds LXR-b.; it is an agonist of ABCA1 that in an experimental study significantly decreased LDL-C and increased HDL, enhancing the expression of the target genes ABCA1/G1, but its development was interrupted because of adverse effects on the central nervous system [67]. Study results have shown that XL-652, a partial LXR agonist with LXR β selectivity, and IMB-808, a potent dual LXRα/β, regulate the gene expression involved in the pathway of metabolizing cholesterol decreasing the lipogenic effect [68]. 

A human study on BMS-852927, a novel LXR β -selective agent, investigated the lipid profile in healthy volunteers. In contrast to the in vitro and non-human primate studies, the study showed a relevant increase in LDL-C and TG with a significant reduction in HDL-C. In addition, the study also showed a significant reduction in neutrophil count [69].

Although more clinical trials are needed to evaluate its efficacy, LXRr agonist could be a promising drug for the treatment of CVD [70].

## 6. PPARβ/δ Agonists

Peroxisome proliferator-activated receptors (PPARs) belong to the group of nuclear hormone receptors and are a subgroup of three ligand-activated transcription factors. PPARs play a crucial role in lipid homeostasis; they regulate many aspects of lipid metabolism. PPARs are proteins activated by the nuclear ligand which, upon activation, modulate transcription of the target genes they control and regulate homeostasis, glucose, triglyceride and lipoprotein metabolism, cell proliferation, vascular tissue function, and inflammation [71]. A (NR1C1), β/δ (NR1C2) and γ (NR1C3) are the three isotypes of PPARs more studied [2].

Among the PPARα agonists, we recognize the fibrates used to treat dyslipidemia; they mainly reduce triglyceride plasma levels, modulate HDL-C and LDL-C, and provide a long-term cardioprotective effect. Thiazolidinediones are potent angplγ agonists used to treat diabetes mellitus; rosiglitazone and pioglitazone are insulin sensitizers used as oral hypoglycemic agents. PPARβ/δ agonists are not yet used in clinical practice, although clinical trials have reported promising results [2].

Recently, studies on potent synthetic PPAR-β/δ agonists, such as GW0742, GW501516, L165041, and GW1929, and their availability in tissue-specific PPAR-β/δ transgenic and gene-targeting mice have resulted in important findings. PPARs have an important role in the regulation of insulin sensitivity, adipogenesis, lipid and energy metabolism, inflammation, and atherosclerosis [71]. PPAR-β/δ (Figure 1) is ubiquitously expressed in animal models, such as mice and rats, and in various human tissues. In mice, PPAR-β/δ protein expression is low in the heart, skeletal muscle, brain, and thymus; intermediate in skin and brown adipose tissue, liver, kidney, lung and vessels; and highest in the gastrointestinal tract. GW501516 is a potent PPAR-β/δ agonist that is approximately 1200 times more selective than other subtypes. 

Treatment with GW501516 in obese and insulin-resistant rhesus monkeys resulted in a dose-dependent increase in plasma HDL-C and a lowering in LDL-C, TG, and insulin levels [71]. In vivo animal models showed that administration of L165041 caused a significant increase of HDL-C and an LDL-C reduction compared to the control group. In addition, treatment of wild-type mice with GW0742 for 14 days caused a modest reduction in serum triglycerides with associated modest increases in serum HDL-C [72]. These studies in animal models stimulated the study of PPAR-β/δ agonist in humans for its potential usefulness in the clinical management of dyslipidemia, especially for treating hypertriglyceridemia [71].

Seladelpar or MBX-8025, a selective PPARβ/δ agonist, is showing promising initial results in the treatment of dyslipidemia. In a randomized parallel study, 181 patients suffering from mixed dyslipidemia were treated for 8 weeks with placebo, atorvastatin 20 mg or seladelpar alone (100 or 50 mg) or in combination with atorvastatin. Seladelpar alone or in combination with atorvastatin reduced LDL-C by 18–43%, non-HDL-C by 18–41%, triglycerides by 26–30%, free fatty acids by 16–28%, and ApoB by 20–38% (*p* < 0.05 for all comparisons). Seladelpar was generally well tolerated, although an unexpected increase in PCSK9 levels was noted during treatment, which requires further investigation [2]. KD-3010 is also a promising PPARβ/δ agonist for the potential anti-atherosclerotic effects, especially in diabetics and obesity patients. A phase 1 clinical trial showed protective and anti-fibrotic effects in preventing liver damage induced by carbon tetrachloride injection (CCl4) or bile duct ligation [73]. CER002 (Cerenis Therapeutics, MI, USA) showed an HDL-inducer in a phase 1 clinical trial and is well-tolerated without significant adverse effects [3]. Selective PPAR-β/δ agonists (HPP593 and a dual agonist PPAR-α/β(δ) (GFT505) are currently being tested in healthy subjects to determine their effect on LDL-C and HDL-C. There are currently no β/δ PPAR targeted drugs available for clinical use in humans, although several selective agonists are at various stages of development. In addition, more efforts should be made to gain information on the post-translational modification of PPARβ/δ and related events. It will be necessary to observe the development of PPARβ/δ agonists as new drugs for the treatment of dysmetabolic diseases in clinical practice [71].

## 7. ANGPTL3 Inhibitors

Angiopoietin-like proteins (ANGPTLs) have emerged as important regulators of lipoprotein metabolism and novel attractive targets for modulating lipid levels and CVD risk. The ANGPTL3 protein (Figure 1), encoded by the ANGPTL3 gene, is located on chromosome 1p31 in humans. ANGPTL3 is a 460 amino acid long protein that belongs to the angiopoietin-like protein family and is secreted exclusively in the liver [74]. It contains an N-terminal heparin-binding domain that increases circulating TG levels by reversibly suppressing the catalytic effect of lipoprotein lipase (ppar), a coiled-coil domain and C-terminal fibrinogen like a domain that could bind to the integrin ανβ3 receptor and facilitate the progression of angiogenesis [75]. A linker region between N- and C-terminal domains function as a furin cleavage site necessary for the ANGPTL3 biological activation [76]. ANGPTL3 is an essential regulator of circulating TG levels through reversible inhibition of the catalytic activity of LPL, an enzyme responsible for the clearance of circulating triglyceride-rich lipoproteins (TRL), chylomicrons, and very low-density lipoproteins (VLDL)—leading to the formation of chylomicron remnants and intermediate-density lipoproteins (IDL), respectively. In addition, ANGPTL3 inhibits the association of LPL with glycosylphosphatidylinositol-anchored HDL-binding protein 1, a protein expressed on capillary endothelial cells that play an essential role in binding LPL to the capillary lumen [77].

Inhibition of ANGPTL3 led to a decrease in plasma triglycerides, ApoB, LDL-C, and HDL-C levels by inhibiting endothelial lipase enzyme activity. In vitro and in vivo studies, have shown that loss of function and inactivation or downregulated expression of ANGPTL3 are associated with a reduction in plasma levels of TGs, LDL-C, HDL-C and consequently the risk of CV events [78]. Pharmacological inhibition of ANGPTL3 could be an attractive therapeutic strategy for lowering lipid levels. Evinacumab, a human monoclonal antibody that inhibits hof3, targeting the C-terminal LPL inhibitory domain of ANGPTL3, has shown promise as a new therapeutic option for unmanageable hypercholesterolemia; the first phase 1 randomized, placebo-controlled, double-blind, clinical trial in human volunteers treated with evinacumab showed a reduction of TG levels up to 76% and LDL-C levels up to 23% [78]. It is well known that statins, cholesterol absorption inhibitors, BA, PCSK9i, and inclisiran lower plasma LDL-C levels by upregulating the expression of LDL-R on hepatocytes. Therefore, their effect is largely dependent on the presence of a functional LDL-R and may be limited in patients with HoFH due to the absence of a fully functional LDL-R [79,80].

The mechanism of ANGPTL3 antagonism, that results in LDL-C reduction is yet to be elucidated, but it is independent of the LDL-R pathway [81,82]. These results suggest that therapeutic antagonism of ANGPTL3 should be effective in lowering LDL cholesterol levels in patients with HoFH and a deficiency in LDL-R-mediated LDL-C uptake. Evinacumab has also been studied in patients with HoFH. In a previous open-label, single-group phase 2 study, the addition of evinacumab to the treatment of 9 HoFH patients resulted in a mean reduction in LDL-C levels of 49% ± 23% at week 4, with an absolute decrease from baseline of (4.06 ± 2.33 mmol/L (157 ± 90 mg/dL). It also reduced apoB, non-HDL-C, triglyceride and HDL-C levels by an average of 46% ± 18%, 49% ± 22%, 47%, and 36% ± 16%, respectively [83]. In a 24-week phase 3 study in which evinacumab was administered to 65 already maximally treated HoFH patients, there was an identical, additional 49% LDL-C reduction compared to placebo, generally with a good tolerance [84]. Evinacumab is therefore an important new therapeutic option for patients with HoFH, especially for those who cannot achieve LDL-C control with current therapies.

Another therapeutic strategy is activating the ANGPTL3 gene in the liver using antisense oligonucleotides. In a phase 1 trial to test the safety and efficacy of ANGPTL3-LRx, 44 participants were randomly assigned to receive a subcutaneous injection of placebo or an antisense oligonucleotide targeting ANGPTL3 mRNA in a single dose (20, 40, or 80 mg) or multiple doses (10, 20, 40, or 60 mg per week for 6 weeks). Statistically, significant lipid reductions were achieved in the multiple-administrations groups in a dose-dependent manner, inducing a maximum of 63% TG reduction and 36.6% reduction in non-HDL-C levels. However, further clinical trials are required to demonstrate its efficacy in CV risk reduction [85,86]. The analysis of ANGPLT3 specific protein structure showed possible additional effects, such as an anti-inflammatory and anti-angiogenic that could increase its anti-atherosclerotic effect. In conclusion, recent studies have revealed genetic and pharmacological ANGPLT3 inhibition’s important role in the modulation of lipoprotein metabolism. Indeed, evinacumab or anti-sense oligonucleotide might promise exciting, beneficial effects on CV risk and high-risk patients’ future outcomes; this will be probably a focus in the research of dyslipidemia in the next years.

## 8. Mipomersen

Mipomersen is an antisense oligonucleotide capable of binding ApoB-100 mRNA to prevent translation of the ApoB protein, the primary apolipoprotein associated with LDL-C [87]. After subcutaneous injection, it is transported to the liver, which causes selective degradation of ApoB-100 mRNA. This leads to decreased production of LDL, VLDL, and Lp(a) [88]. Mipomersen is currently approved by the FDA but not by the EMA, and it is a therapeutic option as an adjunct to lipid-lowering therapy for the treatment of FH in those patients in whom the LDL-C target is not adequately achieved (Figure 2) [89].

Kastelein et al. conducted a phase 1 study to evaluate the tolerability and safety of mipomersen in 36 volunteers with mild dyslipidemia who were randomized to receive either mipomersen at a dose of 50–400 mg per week or placebo. After 4 weeks, mipomersen produced a dose-dependent reduction in ApoB, LDL-C and total cholesterol. The most common ADEs were injection site reactions and increased alanine transaminase levels [90]. Akdim et al. conducted three randomized phase 2 trials to evaluate the efficacy of mipomersen in patients with mild to moderate hypercholesterolemia either as monotherapy or in combination with lipid-lowering agents. All studies showed a significant dose-dependent reduction in LDL-C and ApoB levels, ranging from 7 to 71% and 16 to 71%, respectively. Approximately 50% of participants taking a dose of 400 mg/week experienced transaminase elevations more than three times the upper limit of normal, leading to discontinuation of therapy [91,92,93].

In the phase 3 studies, subjects with moderate to severe hypercholesterolemia on maximal lipid-lowering therapy were randomized to 200 mg mipomersen per week or placebo. After 26 weeks, results showed significant reductions in levels of LDL-C, ApoB, total cholesterol and Lp(a). McGowan et al. showed a significant reduction in LDL-C and ApoB levels in patients with severe hypercholesterolemia on maximal lipid-lowering therapy [94]. Stein EA et al. demonstrated a significant reduction in LDL-C and apoB levels in patients with HeFH on maximally tolerated statin therapy [95].

Thomas G/S et al. reported in non-FH patients at a high risk of CV disease with LDL-C levels ≥ 2.58 mmol/L (100 mg/dL) on maximally tolerated lipid-lowering therapy in the first phase 3 trial, a significant reduction in LDL-C and ApoB levels [96]. In the phase 3 studies, the main side effects were injection site reactions, flu-like symptoms, an increase in liver transaminases, and an increase in liver fat; these results suggest a possible hepatotoxic effect of mipomersen.

Because of the risk of hepatotoxicity, the FDA recommends that baseline levels of transaminases, total bilirubin, and alkaline phosphate be measured, and that levels be monitored at least once a month during the first year of treatment and every 3 months during the second year of treatment. In addition, the FDA requires a risk evaluation and mitigation strategy to manage serious side effects. If clinical symptoms of liver injury are accompanied by a rise in transaminase levels greater than three times the upper limit of normal, treatment with mipomersen should be discontinued. In addition, mipomersen is contraindicated in patients with moderate or severe liver impairment (Child–Pugh class B or C) or active liver disease [87].

## 9. Lomitapide

Patients suffering from HoFH are at high risk of CVD due to elevated LDL-C levels. Conventional lipid-lowering agents often are insufficient in managing this disease, which emphasizes the unmet medical need for potential therapies capable of lowering LDL-C. Novel LDL-R independent drugs are emerging for the treatment of HoFH, including lomitapide [97].

Lomitapide (Figure 2) is a microsomal triglyceride transfer protein (MTP) inhibitor, approved in 2012 in several countries in addition to a low-fat diet and other lipid-lowering drugs or LDL-apheresis to treat patients with HoFH [98]. This new drug acts through an LDL-R independent mechanism targeting the rate-limiting step of lipoproteins synthesis. Specifically, lomitapide prevents lipid transfer by directly binding MTP and blocking its activity in the liver and intestine, thus leading to reduced lipoprotein production, especially of Apo B containing lipoproteins (the hepatic VLDL and the intestinal chylomicrons). Consequently, the levels of plasma lipids lower, including LDL-C levels [99,100].

This mechanism of action would result in the accumulation of VLDL and LDL particles in the liver and lower levels of ApoB-containing lipoprotein in the plasma.

Lomitapide is administered orally, and the liver extensively metabolizes it via the CYP3A4 of which it is a strong inhibitor, thus augmenting the exposure of statins on co-administration [101].

This novel drug, acting through an independent LDL-R pathway, showed significant dose-dependent LDL-C levels reduction when tested on patients with HoFH and low residual LDL-R activity during phase 2 proof-of-concept [102] and phase 3 pivotal trials [103].

Overall, this MTP inhibitor could lower LDL-C levels more than >50% in more than half of HoFH patients, reaching LDL-C targets not achievable with other lipid-lowering therapies alone. These data were affirmed by a phase 3 extension trial (246 weeks), showing that 74% of patients (N = 14) achieved LDL-C goals added to maximally tolerated lipid-lowering therapy [104].

In addition, the potential use of lomitapide in non-HoFH patients with hypercholesterolemia who do not achieve target LDL-C levels despite the use of maximally tolerated doses of other lipid-lowering agents was highlighted.

Samaha FF et al. showed that lomitapide reduced LDL-C levels in a dose-dependent manner even when used as monotherapy: 19% at 5.0 mg, 26% at 7.5 mg, and 30% at 10 mg, each treatment lasting four weeks [105]. In the combined lomitapide plus ezetimibe group, LDL-C was further reduced by 35%, 38%, and 46%, respectively.

Moreover, a post hoc analysis found that patients with HoFH receiving 26 weeks of lomitapide treatment presented no significant differences in LDL-C reduction between apheresis-treated and untreated groups [106]. Inhibition of MTP exhibited a reduction in ApoB-containing lipoprotein from circulation.

Lomitapide’s safety evaluation was monitored over more than 5 years (phase 3 extension trial) and remained under tight monitoring through the LOWER registry [107]. The most significant reported ADEs were: increased risk of hepatotoxicity (elevated levels of transaminases), hepatic fat accumulation, GI-related side-effects (abdominal pain, bloating, constipation, or flatulence), and impairment in fat-soluble vitamins and fatty acids absorption. However, these side effects were generally well tolerated, and most of them disappeared immediately after discontinuation or dose adjustment [103,104,108].

The available data suggest a favorable benefit–risk ratio when the drug is tolerated and support the clinical use of lomitapide for the treatment of patients with HoFH, with monitoring of liver function recommended. Although treatment with lomitapide in non-HoFH patients is an off-label use, lomitapide also has advantages over other agents in certain respects. Further studies are needed to define the safety and efficacy of this novel drug, although research results to date are promising.

## 10. Volanesorsen

Apolipoprotein C-III (ApoC-III) plays an important role in regulating plasma triglyceride levels and is therefore essential for lipoprotein metabolism. It is a component of TRL, which is mainly synthesized in the liver. It is known to inhibit LPL-mediated hydrolysis and impair receptor-mediated hepatic uptake of TRL remnants. At higher concentrations, ApoC-III is associated with both impaired lipolysis and impaired clearance of TRL from the bloodstream because it inhibits the activity of hepatic lipase, an enzyme that acts in the conversion of VLDL to IDL and LDL and in the remodeling of HDL. This leads to the accumulation of atherogenic VLDL and chylomicron remnants.

Volanesorsen (ISIS 304801) (Figure 2) is a 2′-O-(2-methoxyethyl)-modified antisense oligonucleotide specifically targeted to reduce hepatic ApoC-III mRNA [109].

Daniel Gaudet et al. conducted a phase 2, randomized, double-blind, placebo-controlled trial to evaluate volanesorsen in untreated subjects with High TG levels between 3.95 mmol/L (350 mg/dL) and 0.02 mmol/L (2 g/dL) and in patients receiving stable fibrate therapy whose fasting triglyceride levels were between 2.54 mmol/L (225 mg/dL) and 0.02 mmol/L (2 g/dL). Eligible patients were randomly assigned to either placebo or volanesorsen at a dose of 100 to 300 mg once weekly for 13 weeks. The primary outcome was the percentage change from baseline in ApoC-III levels. The mean (±SD) baseline triglyceride levels in the two cohorts were 6.56 ± 3.29 mmol/L (581 ± 291 mg/dL) and 4.25 ± 2.12 mmol/L (376 ± 188 mg/dL), respectively. Volanesorsen administration led to a dose-dependent and sustained reduction in ApoC-III plasma levels, when the drug was administered as a single agent (reduction of 79.6 ± 9.3% in the 300 mg group, 63.8 ± 22.3% in the 200 mg group and 40.0 ± 32.0% in the 100 mg group versus an increase of 4.2 ± 41.7% in the placebo group) and as an add-on to fibrates (decrease of 70.9 ± 13.0% in the 300 mg group and 60.2 ± 12.5% in the 200 mg group versus a decrease of 2.2 ± 25.2% in the placebo group). Concordant reductions of 31.3 to 70.9% were observed in the TG level; no safety concerns were noted. In addition, HDL-C levels increased, and VLDL-C levels decreased in both cohorts in a dose-dependent manner [109].

The study COMPASS (efficacy and safety of volanesorsen in patients with multifactorial chylomicronemia) was a multicenter, double-blind, randomized, placebo-controlled, phase 3 study that aimed to investigate the efficacy and safety of volanesorsen in patients with multifactorial severe hypertriglyceridemia or familial chylomicronemia syndrome (FCS), who had a BMI of 45 kg/m^2^ or less and a fasting plasma TG of 5.65 mmol/L (500 mg/dL) or more [110]. Patients were randomly assigned (2:1) to once-weekly subcutaneous treatment with volanesorsen (300 mg) or placebo for 26 weeks. The primary outcome was the percentage change from baseline to 3 months. Volanesorsen reduced mean plasma TG concentrations from baseline to 3 months by 71.2% versus 0.9% in the placebo group (*p* < 0.0001), corresponding to a mean absolute reduction in fasting plasma Tg of 9.81 mmol/L (869 mg/dL) with volanesorsen versus an increase with placebo of 0.84 mmol/L (74 mg/dL).

The main ADEs were related to tolerability and included injection site reactions (mean 24% of all volanesorsen injections versus 0.2% of placebo injections). Serum sickness occurred in one patient and platelet counts decreased to less than 50,000/μL in one participant in the volanesorsen group [110].

In addition, A. Digenio et al. demonstrated an improvement in metabolic dyslipidemia in patients with type 2 diabetes with a strong relationship between improved insulin sensitivity and suppression of ApoC-III (r = −0.61, *p* = 0.03) and TG (r = −0.68, *p* = 0.01) in plasma [111].

In May 2019, the EMA granted marketing authorization for volanesorsen as an adjunct to diet in adult patients with inadequate response to diet and TG -lowering therapy and with genetically confirmed FCS, who are at high risk for pancreatitis [11].

In summary, the ApoC-III inhibitor ISIS 304,801 has ushered in a new era of lipid-lowering agents that target the TRL and, as non-statin agents, could reduce the progression and consequences of atherosclerosis.

## 11. Conclusions

The increase in CVD in recent years imposes enormous costs in terms of mortality and morbidity [1]. This calls for rapid intervention and aggressive modification of risk factors for atherosclerosis. In particular, lowering LDL-C levels and reducing CVD risk in both primary and secondary prevention is one of the main goals supported in European and American guidelines [10,11].

Statins are the standard therapy for the treatment of hypercholesterolemia. Their efficacy is well established, and their use at the maximum tolerated dose in combination with ezetimibe often allows LDL-C targets to be achieved [3]. However, some patients do not achieve optimal LDL-C goals or do not tolerate statins, especially at high doses. These patients are at high risk of CVD. Therefore, several novel drugs discussed in this narrative review have been introduced to reduce CVD risk, and extensive research is focusing on new molecules to treat hypercholesterolemia (Figure 1, Table 1 and Table 2) [2].

The difficulty in achieving LDL-C targets and reducing CVD risk is even more significant in patients with HoFH [6]. New therapeutic agents have been investigated and introduced into clinical practice to address this difficulty, such as lomitapide and the antisense oligonucleotides mipomersen and volanesorsen (Figure 2, Table 1 and Table 2).

Further clinical trials are needed to provide valuable information on the efficacy of these agents and their role in reducing CVD in patients with dyslipidemia.

## Figures and Tables

**Figure 1 jcm-11-04716-f001:**
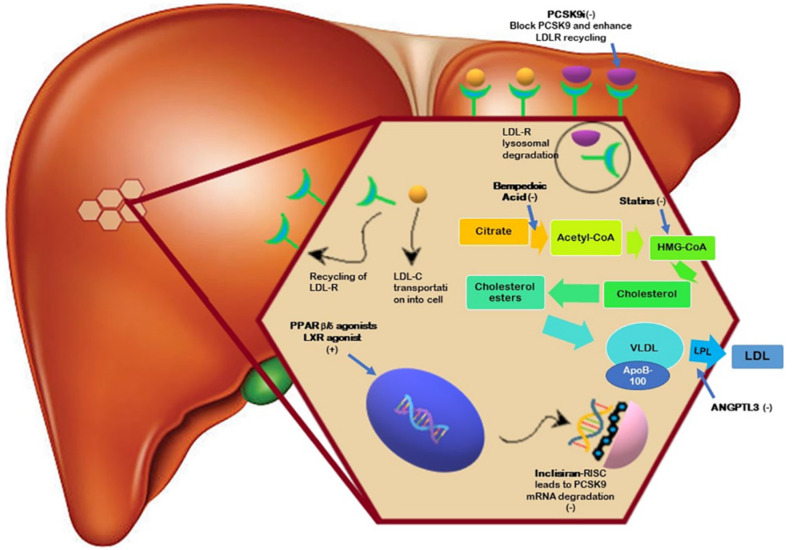
Mechanism of action of novel lipid-lowering drugs used for the treatment of non-familial hypercholesterolemia (non-FH).

**Figure 2 jcm-11-04716-f002:**
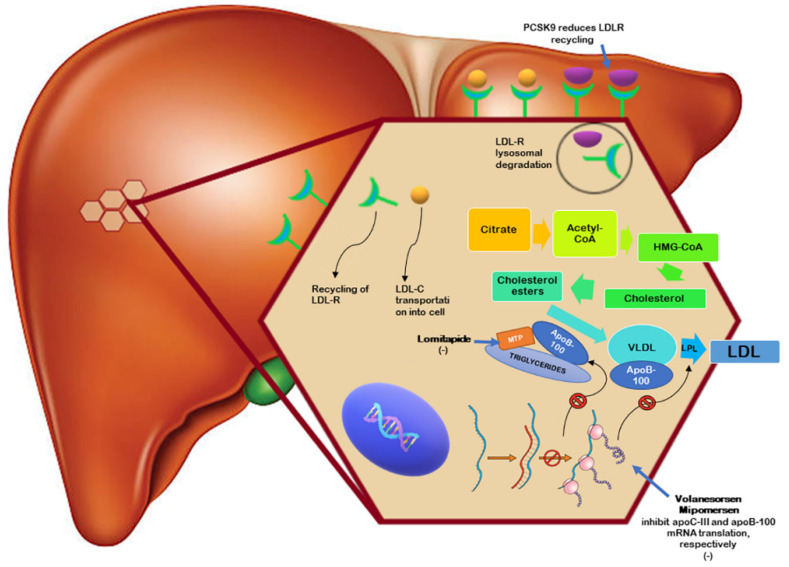
Mechanism of action of novel drugs used in patients with homozygous familial hypercholesterolemia (HoFH).

**Table 1 jcm-11-04716-t001:** Summary of the MOA, main results, and EMA/FDA approval of the novel Lipid Lowering agents.

Drugs	MOA	Efficacy	EMA Approval	FDA Approval
**PCSK9-i**	Monoclonal antibodies that inhibit PCSK9, diminishing the recycling of LDL-R	Medium decrease of LDL-C: 61.9% [12,13]	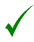	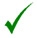
**Bempedoic Acid**	The active metabolite inhibits adenosine triphosphate citrate lyase, leading to reduced acetyl CoA levels in the cholesterol synthesis pathway upstream of HMG-CoA reductase	Medium decrease of LDL-C: 27% [31,32]BA plus ezetimibe decrease LDL-C up to 48% [37]	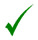	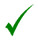
**Inclisiran**	siRNA targeting 3′ UTR of the PCSK9 mRNA leading to suppression of hepatic PCSK9 production	Medium decrease of LDL-C: 52.3% [45]	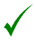	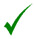
**LXR agonists**	Activation of LXR in macrophages induces cholesterol efflux. LXR reduce cholesterol absorption and promote intestinal excretion, increasing Apo E expression and downregulating Niemann-Pick C1 like 1	No RCTs availableNo human	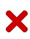	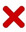
**PPARs agonists**	Proteins that modulate the transcription of the target genes regulating glucose, TG and lipoprotein metabolism, cell proliferation, inflammation, and vascular tissue function	Seladelpar induce a medium decrease of LDL-C by 18–43% [71,72,73]	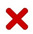	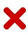
**ANGPTL3 inhibitors**	ANGPTL3 inhibits LPL from combining with the glycosylphosphatidylinositol-anchored HDL binding protein 1: Inhibition of ANGPTL3 results in lower plasma TG, ApoB, LDL-C and HDL-C	Evinacumab induce a medium decrease of TG levels up to 76% and LDL-C levels up to 23% [78]The addition of i.v. evinacumab to the treatment regimen of HoFH patients resulted in a mean reduction of LDL-C levels by 49% ± 23% after 4 weeks [78]	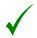	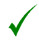
**Mipomersen**	Antisense oligonucleotide capable of binding ApoB-100 mRNA leading to decrease production of LDL, VLDL and Lp(a)	Medium decrease of LDL-C up to 71% [91,92,93]	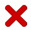	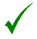
**Lomitapide**	MTP inhibitor leading to reduced lipoprotein production (especially of Apo B)	Used for the treatment of FHMedium decrease of LDL-C levels up to 30% [106]In combination with ezetimibe LDL-C was reduced up to 46%[106]	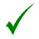	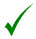
**Volanesorsen**	Antisense oligo-nucleotide target to reduce hepatic ApoC-III mRNA	Medium decrease of TG levels up to 71.2% [110]	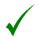	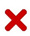

ANGPTL3: angiopoietin-like proteins; BA: Bempedoic acid; EMA: European Medicines Agency; FDA: Food and Drug Administration; FH: familial hypercholesterolemia; HDL-C: high density lipoprotein cholesterol; HoFH: homozygous familial hypercholesterolemia; i.v.: intra venous; LDL-C: low density lipoprotein cholesterol; LPL: lipoprotein lipase; LXR: Liver X Receptors; MOA: mechanism of action; MTP: microsomal triglyceride transfer protein; PCSK9: proprotein convertase subtilisin/kexin type 9; PPAR: peroxisome proliferator activated receptors; RCTs: randomized clinical trials; TG: triglycerides.

**Table 2 jcm-11-04716-t002:** Summary of indications, contraindications, and main side effects of the novel Lipid Lowering agents.

Drugs	Indication	Contraindications	Side Effects
**PCSK9-i**	Adults with primary hyperlipidemia (including HeFH) as an adjunct to diet, alone or in combination with other lipid-lowering therapies; In patients with HoFH as an adjunct to diet and other LDL-lowering therapies (e.g., statins, ezetimibe, LDL apheresis)	Patients with a history of a serious hypersensitivity reaction to PCSK9-i	Nasopharyngitis, upper respiratory tract infection, influenza, back pain, and injection site reactions [20,21]
**Bempedoic Acid**	Adults with HeFH or established ASCVD who require additional lowering of LDL-C as an adjunct to diet and maximally tolerated statin therapy	None	Upper respiratory tract infection, muscle spasms, hyperuricemia, back pain, abdominal pain or discomfort, bronchitis, pain in extremity, anemia and elevated liver enzymes [33]
**Inclisiran**	Adults with HeFH or ASCVD, who require additional lowering of LDL-C as an adjunct to diet and maximally tolerated statin therapy	None	Injection site reaction, arthralgia, urinary tract infection, diarrhea, bronchitis, pain in extremity, and dyspnea [45,46,47]
**LXR agonists**	Not yet approved	No human trials available	No human trials available
**PPARs β/δ agonists**	Not yet approved	No human trials available	No human trials available
**ANGPTL3 inhibitors**	Adult and pediatric patients, aged 12 years and older, with HoFH as an adjunct to other LDL-C lowering therapies	History of serious hypersensitivity reactions to ANGPTL3 inhibitors	Nasopharyngitis, influenza-like illness, dizziness, rhinorrhea, and nausea [85,86]
**Mipomersen**	In patients with HoFH as an adjunct to lipid-lowering medications and diet to reduce LDL-C, ApoB, TC, and non HDL-C	Moderate or severe hepatic impairment, or active liver disease, including unexplained persistent elevations of serum transaminases; Known sensitivity to product components	Injection site reactions, flu-like symptoms, nausea, headache, and elevations in serum transaminases, specifically ALT [87,96]
**Lomitapide**	In patients with HoFH to reduce LDL-C, TC, apo B, and non-HDL-C as an adjunct to a low-fat diet and other lipid-lowering treatments, including LDL apheresis	Pregnancy, concomitant use with strong or moderate CYP3A4 inhibitors, moderate or severe hepatic impairment or active liver disease including unexplained persistent abnormal liver function tests	Diarrhea, nausea, vomiting, dyspepsia, and abdominal pain [103,104,108]
**Volanesorsen**	In adult patients with genetically confirmed FCS and at high risk for pancreatitis, in whom response to diet and triglyceride lowering therapy has been inadequate	Hypersensitivity to the drug; Chronic or unexplained thrombocytopenia. Treatment should not be initiated in patients with platelet count < 1.40 × 10^11^/L	Injection site reactions, serum sickness, and thrombocytopenia [110]

ANGPTL3: angiopoietin-like proteins; ALT: Alanine-Aminotransferase; ApoB: Apolipoprotein B; ASCVD: Atherosclerotic Cardiovascular Disease; CYP3A4: cytochrome P450 3A4; FCS: Familial Chylomicronemia Syndrome; HDL-C: high density lipoprotein cholesterol; HoFH: homozygous familial hypercholesterolemia; LDL-C: low density lipoprotein cholesterol; LXR: Liver X Receptors; PCSK9: proprotein convertase subtilisin/kexin type 9; PPAR: peroxisome proliferator activated receptors; TC: total cholesterol.

## Data Availability

Not applicable.

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
