# Peer review of "Current Options and Future Perspectives in the Treatment of Dyslipidemia"

_jcm, 2022, doi:10.3390/jcm11164716_

Round 1
Reviewer 1 Report
This is a narrative review of available and upcoming pharmacological options for dyslipidemia management. The review provides a fairly systematic and comprehensive take on the topic, with a review of ‘hot’ pharmacological management options for dyslipidemia. I believe the manuscript content covers all relevant aspects of the field, provides specific novel information for the invested reader, and shall prove useful for a plethora of professionals in the field. My major issues relate to form — I believe the manuscript would very much improve with editing and proofreading.
Issues to be addressed:
- Plase, declare this as a narrative review.
- Why the choice of these specific medications? p
- Please provide rationale for choice — i.e., only medications that have been through phase I trials? only medications approved by regulators in either North America or Europe?
- Why did the authors omit omega-3 FA (e.g., icosa pentethyl)?
- While most of the review is focused on LDL-C, TG is mentioned in relation to, and FH thereafter. Maybe try a systematic approach (from dyslipidemia management towards drugs, not the other way around)
- Also, a general paragraph on dyslipidemia diagnosis/categorisation and management (e.g., hypercholesterolemia, including FH, vs. dyslipidemia) would be helpful for the uninformed reader.
- Most of the current approaches have an important role in patients who do not tolerate statin therapy; could you briefly describe the prevalence, possible mechanisms and importance of statin intolerance in this context?
- Also, please define the ‘SI’ acronym when its first introduced (line 86).
- FH is not considered rare — please provide prevalence (so that the reader can appreciate the breadth of the problem).
- Table should be referenced
- What does ‘medium’ LDL reduction in table mean — average? median? Also, where is the data from? (provide references in/for table as well)
- Consider reporting lipid levels in both mg/dL and SI units (mmol/L; the latter is the standardised unit system) — depending on the journal policy
Reviewer 2 Report
This is an interesting well-written overview of current and future options in the treatment of dyslipidemia. It contains both detailed high-level information but is still easy to read and comprehend and has a good structure. It should be interesting to read for the readers of the special issue.
I have some minor concerns:
1. Page 3, Line 107-110: please rephrase the sentence, you should either change the beginning or the part with “…and…”
2. Part 3, BA, page 5, line 193/194:
Ballantyne et al, investigated the association between BA and low-dose statins, in 134 patients randomised to receive BA versus placebo for 12 weeks with baseline LDL-C of 115-220 mg/dL [31].
I do not agree that the association between BA and statins was tested in this article. They compared instead the lipid-lowering efficacy of ETC-1002 versus placebo when added to ongoing statin therapy. Please rephrase
3. Part 4, Inclisiran, page 7, Line 288/289
ORION-4 is a phase III, double-blinded, randomized ongoing controlled trial as-sessing the effects of MACE on participants with pre-existing ASCVD who cannot achieve LDL-C goals.
I do not agree with the expression “the effects of MACE”. MACE is the outcome instead. Please rephrase.
4. Part 5, LXR agonists, page 7, line3 344-346
You quote:
In contrast to the in vitro and non-human primate studies, 340 the study showed a relevant increase in LDL-C and TG with a significant reduction in 341 HDL-C. In addition, the study also showed a significant reduction in neutrophil count. 342 [68] 343
And then summarize:
Despite further clinical studies being required to assess their efficacy, LXRr agonist 344 could be a promising therapeutic agent for the treatment of atherosclerosis in the future 345 [69].
For me, this sounds too positive and promising. Which results do you lean your statement on. For me, the results on the LXRb-selective agents were not that positive and for the non-selective ones you had problems with the side effects. Please explain.
5. Page 10-12, Part 8, Mipomersen
For all the other molecules you have some kind of judgment at the end of the paragraph. How do you think regarding Mipomersen, is it worth to be studied further in spite the toxicity? Does it seem to be promising?
6. Part 9, Lomitapide , page12
Lomitapide (Fig.2) is a microsomal triglyceride transfer protein (MTP) inhibitor, approved in 2012 in several countries in addition to a low-fat diet and other lipid-lowering drugs or LDL-apheresis to treat patients with HoFH [97]. This new drug acts through an LDL-R independent mechanism..
It confuses me if you first name the drug to be approved in 2012 and then call it a new drug. Or are you talking about different molecules?
7. General:
Why did you choose to describe the different molecules and their effects in just this order, is there any explanation for it? If it is you could mention it at the beginning as a kind of introduction to make it easier for the reader to know what to expect.
8. Conclusion:
Did you consider drawing any conclusions comparing the different types of drugs and the results of the research regarding effects and safety up to now? Are some drugs more promising than others? That would be really interesting for the reader. Or is it in your opinion too speculative?
Reviewer 3 Report
It was a well-written manuscript! I suggest authors to shorten it. In addition, I suggest authors to add a new table or to extend table 1 adding indications and contraindications / limitations for every drug.
Round 2
Reviewer 1 Report
Thank you for addressing my comments. This is a well written narrative review that will be very helpful to dedicated professionals.